# Urolithin M5 from the Leaves of *Canarium album* (Lour.) DC. Inhibits Influenza Virus by Targeting Neuraminidase

**DOI:** 10.3390/molecules27175724

**Published:** 2022-09-05

**Authors:** Mengjie Xiao, Fei Cao, Tao Huang, Yun-Sang Tang, Xin Zhao, Pang-Chui Shaw

**Affiliations:** 1School of Life Sciences, The Chinese University of Hong Kong, Hong Kong 999077, China; 2College of Pharmaceutical Sciences, Hebei University, Baoding 077000, China; 3China National Analytical Center, Guangzhou Institute of Analysis, Guangdong Academy of Sciences, Guangzhou 510075, China; 4Li Dak Sum Yip Yio Chin R&D Centre for Chinese Medicine, The Chinese University of Hong Kong, Hong Kong 999077, China

**Keywords:** *Canarium album* (Lour.) DC., influenza virus, NA inhibitor

## Abstract

Ganlanye (GLY), the leaf of *Canarium album* (Lour.) DC., is a traditional Chinese medicinal herb for warm disease treatment. We found that its aqueous extract could inhibit the influenza A virus. To find and characterize anti-influenza virus phytochemicals from GLY, we performed (1) bioassay-guided isolation, (2) a cell and animal assay, and (3) a mechanism study. Bioassay-guided isolation was used to identify the effective components. Influenza virus-infected MDCK cell and BALB/c mouse models were employed to evaluate the anti-influenza virus activities. A MUNANA assay was performed to find the NA inhibitory effect. As a result, urolithin M5 was obtained from the crude extract of GLY. It inhibited influenza virus activities in vitro and in vivo by suppressing the viral NA activity. In the MDCK cell model, urolithin M5 could inhibit an oseltamivir-resistant strain. In a PR8-infected mouse model, 200 mg/kg/d urolithin M5 protected 50% of mice from death and improved lung edema conditions. GLY was recorded as a major traditional herb for warm disease treatment. Our study identified GLY as a potent anti-influenza herb and showed urolithin M5 as the active component. We first report the in vivo activity of urolithin M5 and support the anti-influenza application of GLY.

## 1. Introduction

Influenza is a contagious respiratory disease that induced three pandemics in the last century. They are the “Spanish” flu in 1918, the “Asian” flu in 1957, and the “Hong Kong” H3N2 flu in 1968, causing millions of deaths globally [1]. Seasonal epidemics of influenza threaten public health and socioeconomic in the long run. Some highly pathogenic IAV strains emerge in different geographic regions, like the H7N9 case in southern China in 2013 [2] and the H5N1 case across the globe beginning in 2003 [3]. The aging of the population also contributes to increasing influenza-related morbidity and mortality [4]. Two major classes of anti-flu drugs are M2 ion channel inhibitors, amantadine and rimantadine, and the neuraminidase inhibitor, oseltamivir. However, resistance to amantadine and oseltamivir has occurred because of their wide use [5]. Researchers are trying to find new targets and anti-influenza virus agents.

In traditional Chinese medicine, influenza symptoms are recorded as a warm disease (Wen Bing). Many heat-clearing and detoxicating prescriptions and herbs are used to treat warm disease; for example, Lianhuaqingwen capsule, Jinyinhua and Banlangen [6]. In Fujian Zhong Cao Yao, Ganlanye (GLY), the leaves of *Canarium album* (Lour.) DC. was able to release some influenza-like respiratory symptoms, including a persistent cough, sore throat and asthma [7]. Previously, we isolated three active canaroleosides from GLY with IC_50_ of 11.63~33.30 μM on influenza virus strains, including an oseltamivir-resistant strain. The three phytochemicals target HA or NA proteins [6]. Furthermore, methyl brevifolincarboxylate [8] and isocorilagin [9] from GLY were identified as a PB2 inhibitor and an NA inhibitor. However, these eight chemicals are not highly concentrated in the leaves, and their animal-model-based evidence is lacking. In light of this, we hypothesized that the major active component in GLY was not disclosed. In this study, a high-yield phytochemical urolithin M5 was obtained, and an animal model evaluation was performed, which advanced the anti-flu research of GLY.

## 2. Results

### 2.1. Isolation and Identification of F2-3-4-6

Bioassay-guided isolation was performed to obtain the most active component from the aqueous extract of GLY. After column chromatography, 330 mg F2-3-4-6 was obtained as gray powder, and its structure analysis was performed. Based on the positive HRESIMS data, F2-3-4-6 was found to have the molecular formula of C_13_H_8_O_7_ at m/z 299.01582 [M + Na]+ (calculated for 299.01622). Based on the positive NMR data, lH NMR (CD_3_OD) δ/ppm 8.334 (1H, d, J = 9 Hz), 7.300 (1H, s), 6.754 (1H, d, J = 9 Hz), 13C NMR (CD_3_OD) δ/ppm 161.141, 146.147, 145.676, 143.231, 140.734, 140.239, 132.706, 117.505, 116.896, 111.796. 111.441, 110.973, 107.242, and after comparing the _1_H and _13_C NMR data (Appendix A Appendix A) with those in the literature [10], F2-3-4-6 was identified as urolithin M5. The HPLC spectra showed a purity of 99.8%.

### 2.2. In Vitro Activities of Urolithin M5

The cytotoxicity of urolithin M5 was initially assessed using the MTT assay on MDCK cells. Urolithin M5 had a CC_50_ value of 227.4 μM (Figure 1a). A plaque reduction assay was performed to find the inhibition of urolithin M5 on amantadine-resistant strain A/WSN/33(H1N1) (WSN), oseltamivir-resistant strain A/California/7/2009(H1N1) (pdm09), A/PR/8/34(H1N1) (PR8) and A/Hong Kong/1/68(H3N2) (HK68). As shown in Figure 1b, urolithin M5 suppressed all four strains in a dose-dependent manner. Notably, 25 μM urolithin M5 inhibited all virus growth, indicating that compared with other phytochemicals from GLY, canaroleosides A, B and C, urolithin M5 was the most effective compound, with an IC_50_ ranging from 3.74 to 16.51 μM.

### 2.3. Urolithin M5 Did Not Act on Attachment Stage

To investigate if urolithin M5 exerts an anti-influenza virus activity by interacting with virus attachment, hemagglutination inhibitory assays were performed. In this assay, influenza virus could induce hemagglutinin in chicken erythrocytes by means of forming lattices [11]. This is due to the interaction of the HA1 unit and the sialic acid receptor of the host. Without WSNs, urolithin M5 treatment led to a red spot appearing in the U-bottom reaction well (Appendix A). With WSN, lattice appearance showed in 5–500 μM urolithin M5 or PBS control treatment. PGG, as a positive control, could interrupt hemagglutination and generate erythrocyte red spots. Oseltamivir, as a negative control, did not have an effect on the lattice appearance. These results demonstrated that urolithin M5 had no activity on HA1.

### 2.4. Urolithin M5 Inhibited NA Activities

The effects of urolithin M5 on NA were tested using the NA substrate MUNANA. The reaction of MUNANA and IAV produced a fluorescent signal compound, 4-methylumbelliferone. Urolithin M5 at 10, 100, 125, 250, 500, and 1000 μM was added to the reaction wells, and ten-fold diluted oseltamivir acid was used as a positive control of this assay (except resistant strain pdm09). As shown in Figure 2, urolithin M5 could inhibit NA activity with an IC_50_ of 243.2 μM (WSN), 191.5 μM (pdm), 257.1 μM (PR8), and 174.8 μM (HK68).

### 2.5. Urolithin M5 Improved the Survival Rate of Mice

The in vivo effect of urolithin M5 was examined in a mouse model. Its dose was decided according to reports on in vivo doses of the urolithin family [12,13]. Generally, infected mice lost body weight until day 10, and all died. From the result in Figure 3b, body weights of urolithin M5 and oseltamivir groups rebound at day 9. A total of 200 mg/kg/d urolithin M5 treatment improved the survival rate to 50%, which was significantly higher than the vehicle group (Figure 3a). These results indicated that 200 mg/kg/d urolithin M5 had a protective effect on PR8-infected mice.

### 2.6. Urolithin M5 Reduced Lung Index and Lung Viral Load

Six mice of each group were selected and sacrificed randomly on the fourth day post-infection to obtain their lung index and lung viral titer. Compared with the vehicle group, urolithin M5 treatment decreased the lung index from 1.445 to 0.8875 (Figure 3c), indicating an improvement in lung edema. Lung viral titer was decreased by 0.52 log from Figure 3d, coinciding with the anti-influenza virus effect of urolithin M5 treatment in a cell model.

### 2.7. Urolithin M5 Reduced the Cytokine Expression of Mouse Lung

Influenza virus infection could induce inflammatory storm in the lungs showing from the overexpression of cytokines. To determine the inflammatory markers after urolithin M5 treatment, the right part of the mouse lungs was homogenized on day 4 post-inoculation, and the expression levels of NF-κB, TNF-α and IL-6 were evaluated. As shown in Figure 4a, the three cytokines were upregulated in the vehicle group. Oseltamivir and urolithin M5 treatment both decreased the production of cytokines. Urolithin M5 had better performance than oseltamivir in the regulation of NF-κB levels.

### 2.8. Urolithin M5 Reduced Lung Pathology

The left parts of lung tissues collected on the fourth day post-infection were examined to determine the histopathological changes. As shown in Figure 4b, changes in inflammation were observed after hematoxylin and eosin staining, including interstitial expansion, edema and inflammatory cell infiltration around small vessels. After oseltamivir treatment, exudate around small vessels and bronchus was significantly reduced. After urolithin M5 treatment, fewer bronchi were injured, and fewer inflammatory cells appeared, but inflammatory cell infiltration around small vessels was still observed. These results indicated that urolithin M5 treatment alleviated lung pathology and lesions in influenza virus infection.

## 3. Discussion

After confirming the anti-influenza virus efficacy of GLY crude extract, we performed bioassay-guided isolation and found that urolithin M5 was the most effective high-yield phytochemical, with 330 mg out of 10 kg leaves. Its IC_50_ of 3.74 to 16.51 μM was lower than the three canaroleosides, which had IC_50_ values of 11.63~33.30 μM, according to our previous research [6]. In light of this, in vitro and in vivo activities and mechanism studies of urolithin M5 were carried out. Urolithin M5 could almost suppress A/WSN/33(H1N1), A/PR/8/34(H1N1), A/Hong Kong/1/68(H3N2), and the oseltamivir-resistant strain A/California/7/2009(H1N1). Urolithin M5 has superior performance in the viral-infected mouse model, as shown by the body weights, survival rate, and lung inflammatory situation. The in vivo activities of other reported phytochemicals from this plant, methyl brevifolincarboxylate [8] and isocorilagin [9], are not known. The down-regulatory effect of urolithin M5 on NF-κB, TNF-α and IL-6 was beneficial for the influenza virus-infected mice since excessive pro-inflammatory cytokines/chemokines may cause severe tissue damage after infection [14]. Urolithin M5 has been isolated from *Elaeocarpus* species and has activity against an influenza B strain B/Lee/40 [10,15]. The 50% survival rate at a dose of 200 mg/kg/d and the reduction of lung pathology in mice were firstly reported in this study, and we believe urolithin M5 would be a broad-spectrum and potential anti-influenza agent.

Our study also showed that urolithin M5 acts on NA protein to suppress the viral progeny release, which is similar to the positive control, oseltamivir. However, the binding site of urolithin M5 on NA is unknown. Our further studies will focus on the structure-activities relationship and binding site of the urolithin family, which may help to deal with the resistance of NA inhibitors.

Extracts and phytochemicals from GLY have been reported to have antioxidant activity by scavenging free radicals in a DPPH assay [16], as well as hepatoprotective activity to preserve the liver cell injury induced by CCl4 [17]. The fruits and leaves of GLY are used in health food, pharmaceutical raw materials and skin care products. Our research has broadened the use of GLY as an anti-influenza virus agent.

## 4. Material and Methods

### 4.1. Herb and Reagents

GLY was purchased from Yunnan Labreal Biotechnology Co. Ltd. from Kunming, Yunnan Province, China. The leaves were identified and kept at Li Dak Sum Yip Yio Chin R & D Centre for Chinese Medicine. Minimum essential medium (MEM) and fetal bovine serum (FBS) were purchased from Life Technologies (Gibco, New York, NY, USA). Tolylsulfonyl phenylalanyl chloromethyl ketone (TPCK)-treated trypsin and 4-Methylumbelliferyl-N-acetyl-α-D-neuraminic acid (MUNANA) were purchased from Sigma-Aldrich (St. Louis, MO, USA). The ELISA kit was purchased from BlueGene (Shanghai, China). Thin layer chromatography (TLC) plate, silica gel 60 RP-18 sheets and silica gel 60 sheets were purchased from Merck (Darmstadt, Germany), and a polyamine-6 plate was purchased from Sinopharm (Beijing, China). The stationary phase of column chromatography silica gel was purchased from Qingdao Marine Chemical Inc. (Qingdao, China), silica gel RP-18 was purchased from Merck (Darmstadt, Germany), MCI gel was purchased from Sigma-Aldrich (St. Louis, MO, USA), and Sephadex LH-20 was purchased from GE Healthcare (New York, NY, USA).

### 4.2. Cells and Viruses

Madin–Darby canine kidney (MDCK) cells were purchased from the American Type Culture Collection. Influenza A/WSN/33 (H1N1) (WSN), A/PR/8/34 (H1N1) (PR8), A/HK/8/68 (H3N2) (HK68), and A/California/7/2009 (H1N1) (PDM) were provided by Dr. Chris Ka-Pun Mok (School of Public Health, The Chinese University of Hong Kong) and propagated in MDCK cells. All viral experiments were performed in a Class II biosafety cabinet.

### 4.3. Animal

Specific-pathogen-free female BALB/c mice weighing 16 to 18 g were purchased from Zhejiang Vital River Laboratory Animal Technology Co., Ltd., Guangdong Medical Laboratory Animal Center (Guangzhou, China), and the animal assays were performed at the Guangzhou Institute of Analysis. The animal experiments were carried out with mice provided with a standard laboratory diet and water in a biosafety level 3 laboratory of China National Analytical Center, Guangzhou Institute of Analysis, Guangdong Academy of Sciences (Zhongshan, China) according to the Guidelines of Guangdong Regulation for the Administration of Laboratory Animals. This study was approved by the Animal Experiment Committee at the Guangzhou Institute of Analysis in China, and the approval number was SYXK(Yue) 2019-0201.

### 4.4. GLY Extract Preparation and Bioassay-Guided Isolation

A total of 10 kg GLY was shredded into small pieces before extraction. The leaves were immersed in ten-fold of distilled water and soaked overnight. The leaves were boiled twice for one hour each time. The aqueous extract was collected and concentrated by a rotary evaporator under a vacuum. The concentrated aqueous extract was partitioned with ethyl acetate (EA) and then n-butanol by a separatory funnel to give three different phases. The three different phases were condensed and tested by a plaque reduction assay with WSN strain for their anti-influenza activities. The EA phase (180 g) was found to be the most effective of the three phases. Along each step of isolation, a plaque reduction assay was performed to identify the most effective fraction. The EA phase was loaded to an MCI column and eluted with water, 20%, 40%, 60%, 80% and 95% ethanol gradually to give fractions F1-7. Then, F2 was loaded to a Sephadex LH20 column and eluted with methanol to give sub-fractions F2-1 to F2-4. F2-3 was loaded to a Sephadex LH20 column and eluted with acetone to give sub-fractions F2-3-1 to F2-3-4. F2-3-4 was loaded to a Sephadex LH20 column again and eluted with methanol to give F2-3-4-6 (330 mg). F2-3-4-6 had the best inhibitory effect among all fractions.

### 4.5. Cytotoxicity Assay

The cytotoxicity of F2-3-4-6 was assessed using the 3-(4,5-dimethylthiazol-2-yl)-2,5-diphenyltetrazolium bromide (MTT) assay. Monolayer MDCK cells seeded on a 96-well plate were treated with two-fold diluted F2-3-4-6 in FBS-free MEM. After incubating for 48 h at 37 °C, freshly prepared MTT in PBS was added to each well. After 4 h incubation, the supernatant was removed, and 100 μL of DMSO was added per well to dissolve the formazan crystals. The assay was repeated three times for confirmation. A ClarioStar microplate reader was used to measure the absorbance at 570 nm. CC50 was defined as the concentration that generated a 50% cytotoxic effect.

### 4.6. Virus Infection and Titer

MDCK cells on 6-well plates were washed with PBS. Virus solution A/WSN/33(H1N1) (WSN), A/PR/8/34(H1N1) (PR8), A/California/7/2009(H1N1) (pdm) or A/Hong Kong/1/68(H3N2) (HK68) was 10-fold diluted and added into MDCK cells at 1000 μL per well. After incubating at 37 °C for 2 h, the monolayer cell was coated with FBS-free MEM containing 2 μg/mL TPCK-treated trypsin (unnecessary for WSN infection) and 1% melted agarose in PBS. After infection for 48 h, the agarose plugs were removed, and the cells were stained with a staining solution (0.25% coomassie blue, 10% acetic acid, 50% methanol). The dilution resulting in 200 plaque-forming units (PFU) was determined and used for infection in method 4.7.

### 4.7. Plaque Reduction Assay

MDCK cells were seeded on 6-well plates and incubated overnight. Confluent MDCK cells were washed with PBS and infected with 200 PFU of A/WSN/33(H1N1) (WSN), A/PR/8/34(H1N1) (PR8), A/California/7/2009(H1N1) (pdm) or A/Hong Kong/1/68(H3N2) (HK68). F2-3-4-6 was two-fold diluted with freshly prepared FBS-free MEM containing 2 μg/mL TPCK-treated trypsin (unnecessary for WSN infection). After incubating at 37 °C for 2 h, the monolayer cell was coated with a mixture of F2-3-4-6 solution and 1% melted agarose in PBS. After infection for 48 h, the agarose plugs were removed, and the cells were stained with a staining solution (0.25% coomassie blue, 10% acetic acid, 50% methanol). The assay was repeated three times for confirmation. The concentration of F2-3-4-6 that inhibited 50% of virus-induced plague was determined as the IC_50_ (50% inhibitory concentration).

### 4.8. HMR and HPLC Analysis of F2-3-4-6

NMR spectra were recorded on Bruker Avance 500 MHz (^1^H NMR) and 125 MHz spectrometers (^13^C NMR). DMSO-d6 was used as the solvent. Chemical shifts (δ) are reported as part per million (ppm). High-resolution mass spectra (HRMS) were assessed on an Agilent 6210 LC/MSD TOF spectrograph. High-performance liquid chromatography (HPLC) was performed on a Shimadzu LC-20AT Series HPLC using an Anglient OD-H column eluted with a mixture of ACN and H_2_O (5:95). All spectra are listed in the Appendix A.

### 4.9. Hemagglutination Inhibitory Assay

A hemagglutination inhibitory assay was performed with erythrocyte and WSN virus. WSN (4HA units) and two-fold diluted F2-3-4-6 were added to a U-bottom plate and mixed slightly. After incubating for 30 min at 37 °C, 0.05% chicken erythrocytes in PBS were freshly prepared and added to the reaction wells. After incubating for 30 min at room temperature, hemagglutination could be observed. Pentagalloyglucose (PGG) was used as a positive control, and oseltamivir acid was used as a negative control. The reaction well without virus solution was included to confirm the effect of compounds on chicken erythrocytes. Three parallels were performed for confirmation.

### 4.10. Neuraminidase Inhibitory Assay

To confirm the inhibitory effect of F2-3-4-6 on influenza virus NA, MUNANA and four different viral strains were used to perform an NA inhibitory assay [8]. MES buffer and MUNANA were prepared, respectively. Each reaction well of a black 96-well plate (Costar) contained 30 mM MES buffer, virus solution, F2-3-4-6 dilution and distilled water to make up the volume of 90 μL. After incubating at 37 °C for 30 min, 10 μL 1 mM MUNANA solution was added immediately to each well. After 30 min at 37 °C, the fluorescence intensity (F) of each well was measured by a CLARIOstar multi-mode microplate reader with an excitation wavelength of 322 nm and an emission wavelength of 450 nm. Oseltamivir acid was used as positive control, and three parallels were performed for confirmation. The inhibitory effect of F2-3-4-6 on NA activity was calculated by:NA inhibition (%) = (Fcontrol − Fcompound)/(Fcontrol − Fblank) × 100%

### 4.11. Anti-Influenza Virus Test in Mouse Model

Mice were divided randomly into four groups, with 12 mice per group. The F2-3-4-6, oseltamivir, and vehicle groups were intranasally infected with 2 MLD50 of PR8 in a volume of 20 μL. The F2-3-4-6 groups were orally administered 200 mg/kg/day of F2-3-4-6. The oseltamivir group were orally administered oseltamivir at 65 mg/kg/d as a positive control. The control and vehicle groups were treated with 0.5% CMC solution only. Herb treatment was administered once a day for six consecutive days. Body weights were recorded for 16 consecutive days.

### 4.12. Sample Collection and Detection

On the fourth day post-inoculation, three mice of each group were randomly selected and sacrificed for lung tissue collection. Lung tissues were obtained and weighed. The left part of the lung was stained with formalin for histopathological observation. The right part of the lung was homogenized in MEM containing antibiotics (0.1% penicillin-streptomycin) with a homogenizer three times for 20 s at 4 °C. Homogenates were centrifuged at 12,000 rpm for 5 min at 4 °C. Supernatant was aliquoted to obtain the lung viral load and NF-κB, TNF-α and IL-6 expression. The lung index was calculated as follows:Lung index = (lung weight/body weight) × 100%

### 4.13. Lung Viral Load and NF-κB, TNF-α and IL-6 Expression Analysis

Lung viral load was determined with a plaque-forming assay. MDCK cells were seeded on a 24-well plate and incubated overnight. Lung homogenates were ten-fold diluted and infected with MDCK cells. After incubating for 2 h, the medium was replaced with a mixture of 1% boiling agarose in PBS and FBS-free MEM containing 2 μg/mL TPCK-treated trypsin. After 48 h, agarose plugs were removed, and cells were stained with staining solution (0.25% Coomassie blue, 10% acetic acid, 50% methanol). Plaque was counted to obtain the PFU of each lung. NF-κB, TNF-α and IL-6 expression of lung homogenate was determined by an ELISA kit according to the manufacturer’s protocol. The optical density at 450 nm of each reaction well was read by a CLARIOstar multi-mode microplate reader (BMG Labtech, Offenburg, Germany).

### 4.14. Histopathological Observation

The left parts of lung tissues were processed for paraffin embedding and cut into 4-μm-thick sections. Tissue sections were stained with hematoxylin and eosin to observe histopathological alterations.

### 4.15. Statistical Analysis

Graphpad Prism 6.0 (Graphpad, San Diego, CA, USA) was used for all statistical analyses, and data were calculated as mean values of three independent experiments. For the multiple-group comparison, a one-way ANOVA was used. *p* < 0.05 was regarded as statistically significant (* *p* 0.05, ** *p* 0.01, *** *p* 0.001).

## 5. Conclusions

Ganlanye, the leaves of *Canarium album* (Lour.) DC., were recorded as a major traditional herb for warm disease treatment. In our study, urolithin M5 was identified as a potent anti-influenza virus agent from this herb. Our research provides scientific evidence for the application of GLY in influenza treatment and improves public confidence to traditional medicine.

## Figures and Tables

**Figure 1 molecules-27-05724-f001:**
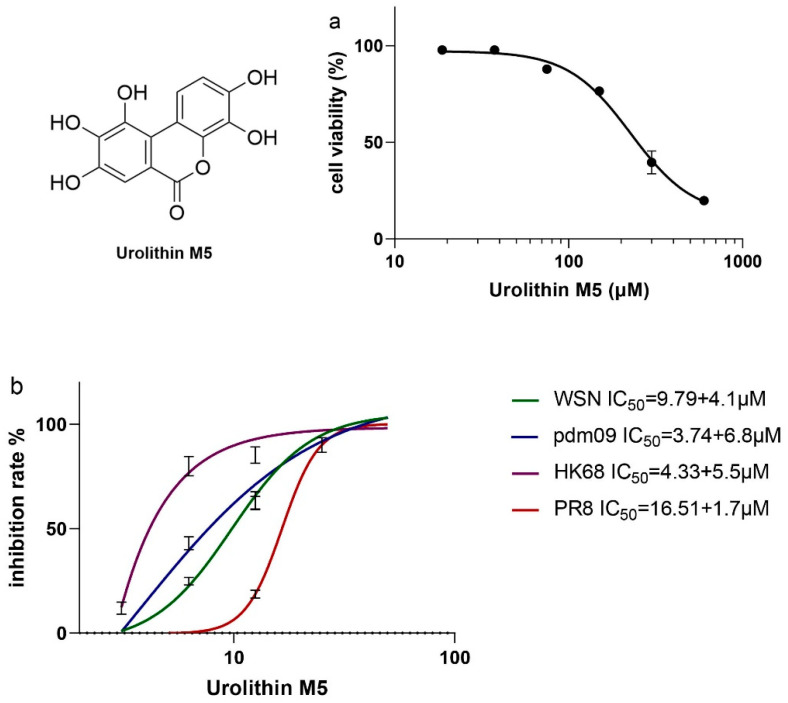
In vitro activities of urolithin M5 were identified by plaque reduction assay. Urolithin M5 inhibited four IAV strains in a dose-dependent manner, and CC_50_ (**a**) and IC_50_ (**b**) values were calculated by Graphpad Prism 6.0.

**Figure 2 molecules-27-05724-f002:**
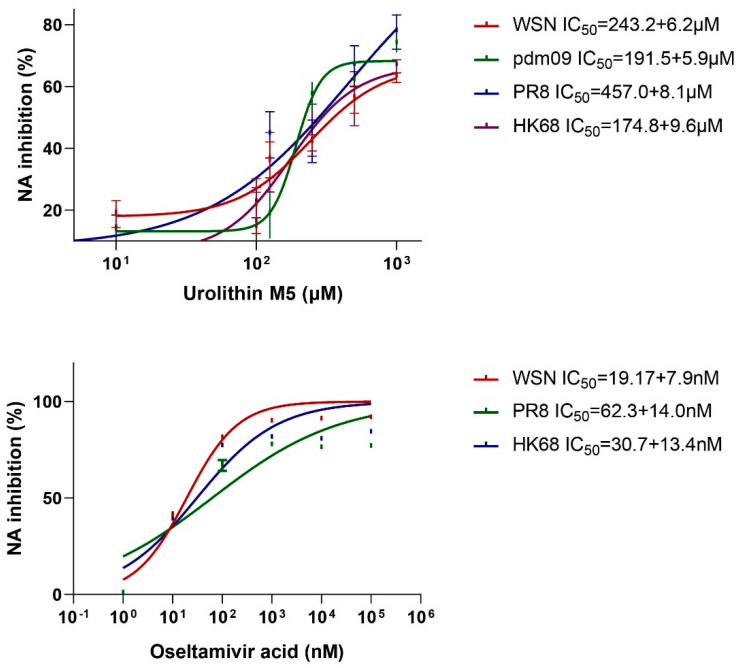
Assay of urolithin M5 on NA inhibition. Oseltamivir acid was used as positive control with an IC_50_ of 19.17 nM (WSN), 62.3 nM (PR8) and 30.7 nM (HK68). The experiments were repeated three times for confirmation.

**Figure 3 molecules-27-05724-f003:**
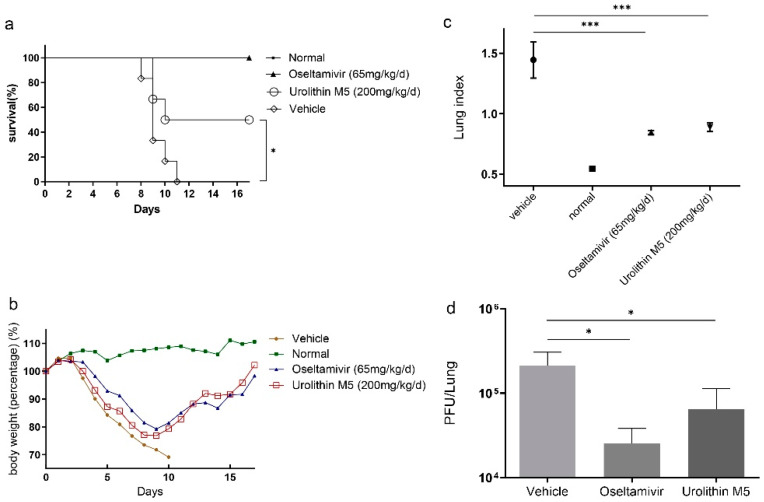
Urolithin M5 exhibited antiviral activities in PR8-infected mice. PR8-infected mice were administrated orally with 200 μL of 200 mg/kg/d urolithin M5, 65 mg/kg/day oseltamivir or 0.5% CMC for 6 consecutive days. (**a**) Body weights and (**b**) survival rate are shown. The efficacy of urolithin M5 and oseltamivir treatment on (**c**) lung index and (**d**) lung viral load of mice are shown. Six mice of each group were measured, and each assay was performed three times for confirmation. (* *p* 0.05, *** *p* 0.001).

**Figure 4 molecules-27-05724-f004:**
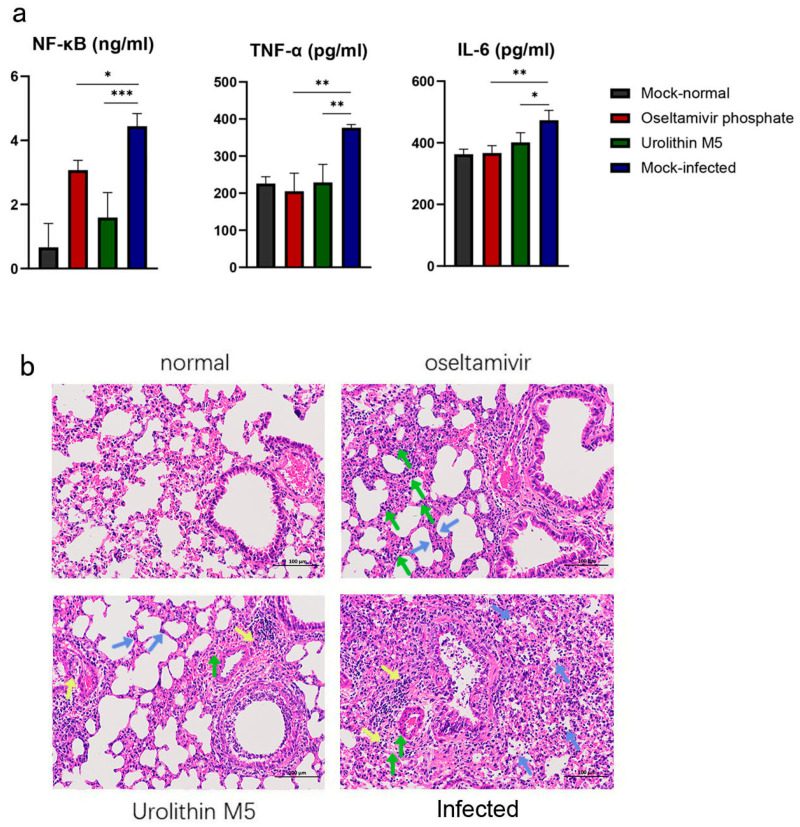
(**a**) Effect of urolithin M5 treatment on NF-κB, TNF-α and IL-6 levels using ELISA assay. (**b**) Histopathologic changes of mice from the normal group, oseltamivir group, urolithin M5 group, and vehicle group on the fourth day. Inflammatory changes are labeled as interstitial expansion (green arrow), small vessel inflammatory cell infiltration (yellow arrow), and alveolar lumen inflammatory cell infiltration (blue arrow) (magnification: 200×). (* *p* 0.05, ** *p* 0.01, *** *p* 0.001).

## Data Availability

Not applicable.

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
