# Peer review of "Urolithin M5 from the Leaves of Canarium album (Lour.) DC. Inhibits Influenza Virus by Targeting Neuraminidase"

_molecules, 2022, doi:10.3390/molecules27175724_

Round 1
Reviewer 1 Report
Review of Manuscript # molecules-1888599
In the manuscript entitled “Urolithin M5 from the leaves of Canarium album (Lour.) DC. inhibits influenza virus by targeting neuraminidase”, the authors extracted a new chemical, urolithin M5, from the leaves of Canarium album to show its inhibition capacity on IAV replication. They first identified the molecular formula of urolithin M5 and tested it in vitro to show that urolithin M5 inhibits NA activities rather than HA activities. The in vivo experiments showed more mice survived with urolithin M5 treatment after the PR8 challenge, and urolithin M5 could lower the cytokine release from the lung cells. Overall, this well-written paper presents a compelling set of data to support urolithin M5 as a potential anti-influenza virus drug, but it fails to prove urolithin M5 can be useful to oseltamivir-resistant strains. In conclusion, this paper should be accepted with revision or rephrase of the introduction.
Major comment:
- As one of the purposes of finding new substrates to inhibit IAV is to overcome the resistance of amantadine and oseltamivir, one would expect to see if urolithin M5 will work better than oseltamivir in oseltamivir-resistant strain. Is it possible to infect the mice with an oseltamivir-resistant strain and treat the mice with urolithin M5 to see if urolithin M5-treated mice survive better? Dr. Guy Boivin has published a paper with an oseltamivir-resistant Victoria (H3N2) strain (PMID: 25114143).
Minor comment:
- Please include in vitro IAV infection protocol in the method section.
- Please put Oseltamivir treated group (the negative control) in Fig. S5 for a better understanding of the data.
- Labeling mistake. The concentration of oseltamivir in Fig. 2. is not correct. The diagram labels uM while the legends label nM.
- Cytokine determined by ELISA should be phrased as produced not expressed unless RT-PCR data is showing the transcription level goes down with urolithin M5 treatment. Please rephrase or show the RT-PCR data.
Author Response
In the manuscript entitled “Urolithin M5 from the leaves of Canarium album (Lour.) DC. inhibits influenza virus by targeting neuraminidase”, the authors extracted a new chemical, urolithin M5, from the leaves of Canarium album to show its inhibition capacity on IAV replication. They first identified the molecular formula of urolithin M5 and tested it in vitro to show that urolithin M5 inhibits NA activities rather than HA activities. The in vivo experiments showed more mice survived with urolithin M5 treatment after the PR8 challenge, and urolithin M5 could lower the cytokine release from the lung cells. Overall, this well-written paper presents a compelling set of data to support urolithin M5 as a potential anti-influenza virus drug, but it fails to prove urolithin M5 can be useful to oseltamivir-resistant strains. In conclusion, this paper should be accepted with revision or rephrase of the introduction.
Major comment:
- As one of the purposes of finding new substrates to inhibit IAV is to overcome the resistance of amantadine and oseltamivir, one would expect to see if urolithin M5 will work better than oseltamivir in oseltamivir-resistant strain. Is it possible to infect the mice with an oseltamivir-resistant strain and treat the mice with urolithin M5 to see if urolithin M5-treated mice survive better? Dr. Guy Boivin has published a paper with an oseltamivir-resistant Victoria (H3N2) strain (PMID: 25114143).
In vivo assay of oseltamivir-resistant strain could explain its effect more clearly. However, because of its drug resistance, stricter biosafety equipment are needed. Therefore, we chose a widely-used strain PR8 for animal model.
Minor comment:
- Please include in vitro IAV infection protocol in the method section.
- Please put Oseltamivir treated group (the negative control) in Fig. S5 for a better understanding of the data.
- Labeling mistake. The concentration of oseltamivir in Fig. 2. is not correct. The diagram labels uM while the legends label nM.
- Cytokine determined by ELISA should be phrased as produced not expressed unless RT-PCR data is showing the transcription level goes down with urolithin M5 treatment. Please rephrase or show the RT-PCR data.
Changes have been made based on these comments. In vitro IAV infection protocol was included as method 4.6.
Reviewer 2 Report
Xiao et al investigated the role of urolithin M5 to inhibit influenza neuraminidase. Although the study has potential scientific rigor, still some major needs to be added.
The comments are below:
The main research question is missing from the manuscript. The hypothesis and the perspectives are not cleared in the intro. To draw possible conclusion, the hypothesis need to be justified. However, no specific info is added in the manuscript.
Were the animal need to be sacrificed? If so, what is the way for the euthanasia?
How the partitioning of the solvents done? Was it followed any specific method?
Line 123: 37C need to be corrected.
The equation is misleading. Its need to be changed accordingly.
The authors should be more careful about the figure and main text. In figure 4, TNF-alpha is written as TNFa, although in main text, they used the Greek letter. It is suggested to be uniform. In addition, what is indicated by the histopathological changes are not that lucid. It is suggested to add bar graph for inflammation score. Also, need to use arrow if feasible.
Discussion need more elaborated description of the published data and the authors claim.
Author Response
The main research question is missing from the manuscript. The hypothesis and the perspectives are not cleared in the intro. To draw possible conclusion, the hypothesis need to be justified. However, no specific info is added in the manuscript.
Relevant description was improved.
Were the animal need to be sacrificed? If so, what is the way for the euthanasia?
All survived mice were executed by neck dislocation.
How the partitioning of the solvents done? Was it followed any specific method?
The partition was performed by separatory funnel.
Line 123: 37C need to be corrected.
Has been corrected.
The equation is misleading. Its need to be changed accordingly.
The equation was changed as Lung index = (lung weight/body weight)×100%
The authors should be more careful about the figure and main text. In figure 4, TNF-alpha is written as TNFa, although in main text, they used the Greek letter. It is suggested to be uniform. In addition, what is indicated by the histopathological changes are not that lucid. It is suggested to add bar graph for inflammation score. Also, need to use arrow if feasible.
Figure 4 has been revised but we could not get an exact inflammation score because of limited number of samples.
Discussion need more elaborated description of the published data and the authors claim.
The discussion has been improved.
Round 2
Reviewer 2 Report
The authors changed the manuscript accordingly. It's now can be published.